# A Novel Remote Visual Inspection System for Bridge Predictive Maintenance

Alessandro Galdelli [1,*,†]  , Mariapaola D'Imperio [2,†]  , Gabriele Marchello [2], Adriano Mancini [1]  , Massimiliano Scaccia [2], Michele Sasso [3], Emanuele Frontoni [4]   and Ferdinando Cannella [2]

1  VRAI Lab, Dipartimento di Ingegneria dell'Informazione, Università Politecnica delle Marche, 60131 Ancona, Italy; a.mancini@univpm.it
2  Industrial Robotics Facility, Istituto Italiano di Tecnologia, 16163 Genoa, Italy; mariapaola.dimperio@iit.it (M.D.); gabriele.marchello@iit.it (G.M.); massimiliano.scaccia@iit.it (M.S.); ferdinando.cannella@iit.it (F.C.)
3  Ubisive Srl, 62012 Civitanova Marche, Italy; michele.sasso@ubisive.it
4  Department of Political Sciences, Communication and International Relations, University of Macerata, 62100 Macerata, Italy; e.frontoni@unimc.it
*  Correspondence: a.galdelli@univpm.it
†  These authors contributed equally to this work.

**Abstract:** Predictive maintenance on infrastructures is currently a hot topic. Its importance is proportional to the damages resulting from the collapse of the infrastructure. Bridges, dams and tunnels are placed on top on the scale of severity of potential damages due to the fact that they can cause loss of lives. Traditional inspection methods are not objective, tied to the inspector's experience and require human presence on site. To overpass the limits of the current technologies and methods, the authors of this paper developed a unique new concept: a remote visual inspection system to perform predictive maintenance on infrastructures such as bridges. This is based on the fusion between advanced robotic technologies and the Automated Visual Inspection that guarantees objective results, high-level of safety and low processing time of the results.

**Keywords:** bridge monitoring; visual inspection; multi-camera system; autonomous robots; artificial intelligence



## 1. Introduction

A bridge is an infrastructure built to overpass a physical obstacle, such as rivers or valleys. It was, it is and it will be a key element in the development of humankind; thus, nowadays there are millions of bridges around the world, which all need maintenance to prevent collapse [1]. The key point for preventing a disaster is predictive maintenance. Unfortunately, due to the large number of bridges around the world, the costs of the maintenance operations are very high. In the case maintenance is performed too early with respect to the infrastructure lifespan, the risk is to waste money and resources. Conversely, in the case of late maintenance, disaster can strike. Consequently, it is clear that the solution is a trade off between costs and safety [2–4]. One of the latest disaster in this field is the one happened in Genova in 2018, sadly known for the Morandi Bridge collapse that took 43 lives away. After this tragedy, "why?" and especially "can this happen again?" were the two questions that started spreading the most, frightening the Italian people; while the former was matter of justice (and at the time this paper is being written, this is still under investigation), the latter was immediately addressed by researchers and companies in order to find reliable solutions.

In general, an inspector conducting a bridge inspection aims at detecting the presence of flaws, defects or potential problem areas that may require maintenance. Conventionally, it is done by bringing the inspector as close as possible to the bridge by using snooper trucks (Figure 1a) or a man lift (Figure 1b).

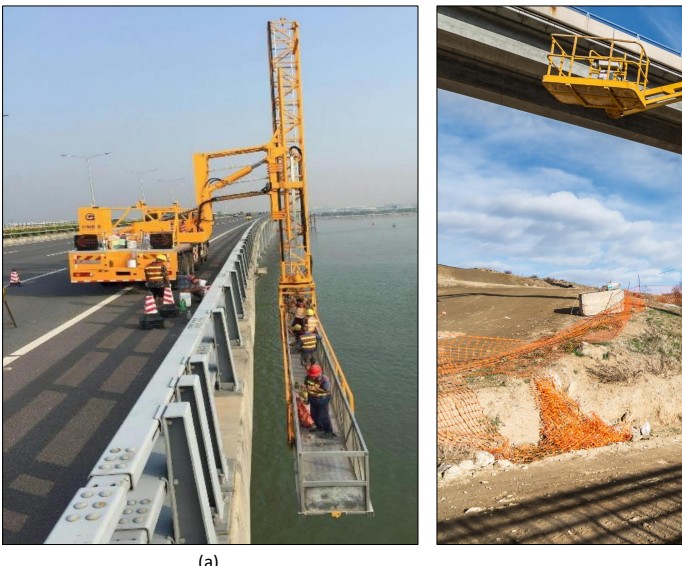
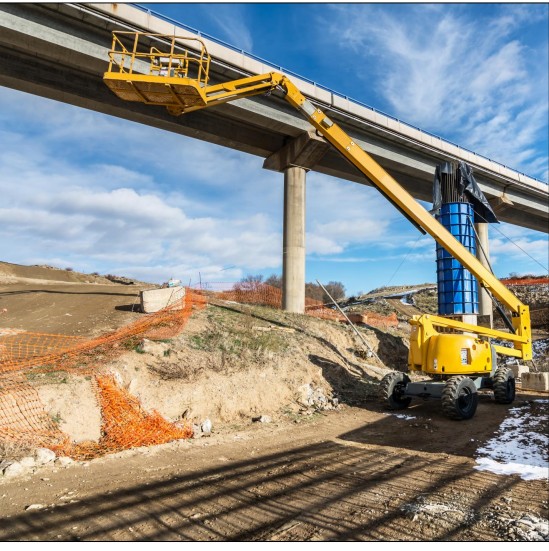

(a)  (b)

**Figure 1.** (**a**) Snooper Truck (Source: [5]. Reprinted with permission from Ref. [5]. Copyright 2021 MDPI); (**b**) Man Lift.

The main limits/challenges posed by the traditional methods are related to the non-objective results of the inspection, as they are heavily influenced by the inspector experience [6]. In addition, at least two people are needed to perform the inspection; a portion of the bridge has to be closed to place the snooper truck; the locations to inspect are not always easily reachable, and therefore, guaranteeing the safety of the human requires very high costs [7]. Consequently, researchers have to propose solutions able to tackle all the aforementioned limits [8]. Such solutions aim (i) to reduce the risks that a human faces while suspended in the void many meters high, (ii) to reach narrow or hidden locations, (iii) to avoid the huge amount of logistical planning for the inspection and the bridge closure, and (iv) to obtain an inspection that is objective, automatic and repeatable [9]. Two of the most investigated fields focus on the substitution of the human on site, and the objectification and repeatability of the inspection process. Several solutions have been proposed to tackle the former problem, such as recurring to mobile inspection robots [9] or Unmanned Aerial Vehicles [10,11]. However, the presence of the operator on site continues to play a fundamental role in the inspection task (e.g., pilot, co-pilot, expert in the inspection domain to supervise the flight also stating the area of interest), and hence, it still cannot be replaced. Furthermore, in order to obtain objective and repeatable analyses, it is fundamental to remove the dependency of the inspection results from the inspector experience, which unfortunately highly depends on physical aspects and the operator set of skills [8]. However, the latest progress in the field of vision systems enabled a significant leap towards the realization of support systems for the inspector, able to perform highly safe, fast, and objective inspections [12].

Automated Visual Inspection (AVI) is a crucial operation in monitoring and inspecting infrastructures (e.g., bridges), widely adopted to guarantee a high level of safety and requiring low processing time. In order to design a methodology and to select the hardware components necessary to perform AVI of bridges, an analysis of the state of the art was conducted. The bridge inspection has to take place under various weather conditions and during the night/day cycle, hence accounting for a wide spectrum of visibility conditions. In order to meet these requirements, a system capable of performing a complete inspection and detecting even micro flaws has to be developed. Consequently, a multi-technology vision system using three different cameras is proposed. The adoption of RGB digital cameras enabled the development of three different inspection methodologies: (i) raw image inspection, (ii) image enhancement or (iii) autonomous image processing [13,14]. In the case the analysis is based on raw images, the inspector manually processes the whole

collection of recorded images. Unfortunately, such an analysis is often time consuming given the amount of data, and prone to inaccuracies or human errors. Thus, in order to overcome these problems, image enhancement algorithms have been used to improve the quality of the images, hence facilitating the identification of cracks. This task is typically accomplished by using edge detection algorithms (e.g., Canny, Sobel, or Gaussian), segmentation techniques (e.g., thresholding or Watershed), percolation algorithms and filtering operations [12,15–17]. Consequently, the application of these techniques has the great advantage of mitigating the inspector fatigue. Conversely, autonomous image processing can be performed via Artificial Intelligence (AI), in detail by applying Machine Learning (ML) or Deep Learning (DL) techniques, which enable the automatic detection of flaws [18–21].

In the recent years, other types of cameras were developed and found potential application in AVI. For instance, the multispectral cameras that are able to capture image data at specific frequencies across the ultraviolet (UV)/visible (Vis)/near infrared (NIR)/short-wave infrared (SWIR) electromagnetic spectrum, covering the 175 nm to 2500 nm wavelength range. The information at different wavelengths may be separated by applying filters, or using instruments that are sensitive to specific values. Hence, it is possible to retrieve image information out of frequencies beyond visible sight (such as infrared), which the human eye fails to capture [22,23]. Different types of multispectral systems have been developed over time, based mainly on three techniques: (i) multiple cameras with different spectral range to a machine vision setup pointing at the target; (ii) filter wheel camera capturing multi-channel spectral images by rotating filters in a filter wheel mounted in front of the sensor or the lens; (iii) single sensor camera using Bayer Color Filter Array (CFA) and demosaicing/debayering. The latter acquisition method is also known as instantaneous mosaic imaging. Multispectral imaging has been employed in various applications (such as agriculture [24–26], medicine [27] or industries [28,29]) by using machine vision cameras to improve the inspection capabilities.

Furthermore, depth cameras, named RGB-D, have been widely used recently. These devices transmit the three standard RGB channels along with depth information (related to the distance to the sensor) on a per-pixel basis. Nowadays, this type of device is becoming increasingly popular due to their capability of reconstructing in 3D the imaged scenes at an affordable cost and in real time. These features make RGB-D cameras ideal for capturing the 3D information needed to perform a bridge inspection [30,31]. Having a highly accurate model in 3D of the cracks would enable their evaluation and quantification. Hence, several contactless and nondestructive techniques have been developed [32]. The Close-Range Photogrammetry (CRP) is a notable technique to obtain a point cloud by using passive sensors, which enable 3D measurements and drawings. Among other nondestructive techniques, Terrestrial Laser Scanning (TLS) and photogrammetry are used to reduce the time required to obtain all the geometric data needed to build a detailed 3D finite element model [33,34]. All of these technologies implement dense point cloud and image-based inspections that can be manned, unmanned or automated [13,35–38]. Typically, manned inspections are costly, time-consuming and in some cases inaccurate. Conversely, unmanned or automated inspections reduce implementation costs and speed up the image analysis phase.

In this scenario, the main objective of this work is to obtain a model of the bridge that is accurate in all the details, and as close as possible to reality, by applying different vision technologies. The combined use of these technologies improves the detection of various types of damage, and increases the reliability of assessment results. Nonetheless, the integration of multiple technologies within a single system, and thus, the synchronization of hardware and data, is still an open challenge and a significant effort is necessary in order to design effective integration strategies. For these reasons, the authors of this paper proposed a solution that integrates robotics and vision to support an automated inspection. The first proof of concept (*p.o.c.*) implementing the proposed solution was installed in the new San Giorgio's bridge in Genova. It has a monitoring system composed of 240 fixed sensors for "standard measurements" of the structure parameters, and, above all, two

robots (one for the north carriageway and the other for the south carriageway of the bridge, respectively) named Robot Inspection Units (RIUs) for "non-standard measurement". RIU is equipped with sensors able to scan the lower surface of the deck to detect potential anomalies. Moreover, the designed system has to perform the acquisition and analysis of the pictures taken by the vision technologies in less than 15s, a time limit set by the manufacturer in order to balance processing time and quality of the inspection. These robots have been in charge of inspecting San Giorgio's bridge since 2021. RIU is capable of autonomously performing several tasks:

- Inspecting the lower surface of the deck;
- Collecting and processing the data to find any anomalies;
- Interrupting operations and safely reaching their charging stations in the event of adverse environmental conditions.

Moreover, the design of the robot accounts for further implementations of novel technologies in order to improve the anomalies' detection and the scanning of the surfaces with multispectral images.

In this paper, the main functions of the robots and the results of the preliminary tests are described. Furthermore, speculations on the future enhancements of the Robot Inspection (RI) and its vision system are discussed at the end of the paper, depending on the versatility of the robot.

## 2. Materials and Methods

The system architecture presented in this paper is composed of two main functional groups: (i) a multi-camera system (MCS) and (ii) a robotic platform RIU. These two groups work in parallel, performing autonomous flaws detection, damage quantification and producing a report of the results (Figure 2). In particular, the vision system installed on RIU was responsible for taking pictures, while RIU moved along the bridge. Moreover, the RIU platform was designed accounting for multiple replicas of the MCS, hence reducing the time needed to scan the bridge. In fact, using several MCSs lowers also the time required by the robot to cover all the locations of the bridge with the vision system. The RIU could be operated trough three main control modes, such as:

A Commands from Human–Machine Interface (HMI) application of the PC Robot Server in the control room (see Section 2.4);
B Local control on the robots themselves via push buttons;
C Controls from remote control room via Supervisory Control and Data Acquisition (SCADA) (see Section 2.4).

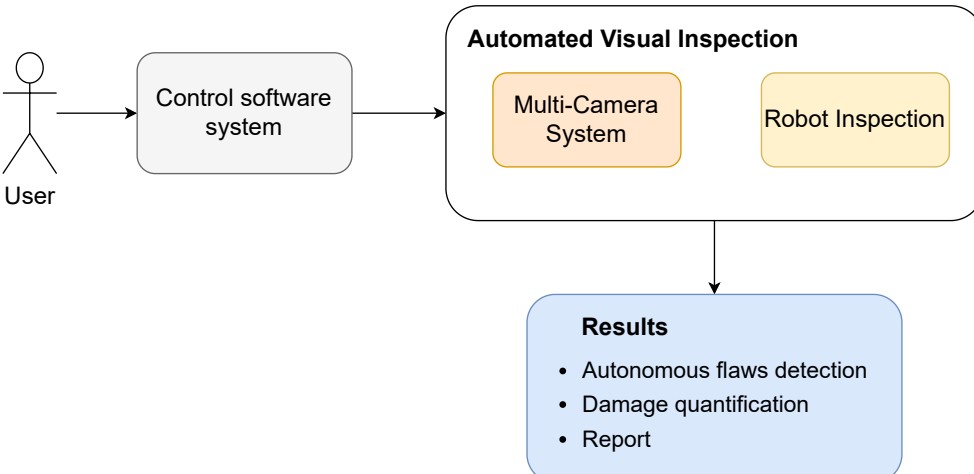

**Figure 2.** Schematic representation of the flowchart modeling the system architecture.

The inspection routine includes RI scanning the underside of the bridge, by starting from the base (the origin point of the relative coordinates frame) and moving towards the end of the bridge. During the longitudinal movement of the robot along the $z$ axis, a retractable beam belonging to the robot is activated along the $x$ axis. As a consequence, the cameras acquire all the sub-areas of the bridge, thus producing a complete scan of a slice of the lower surface of the deck. The cameras activity is synchronised with the movement of the robot arm, which is triggered by the completion of the acquisition cycles of all the MCSs. As soon as the robot reaches the end of the bridge in the $z$ direction (i.e., the terminal point), the inspection is considered as completed. As a result, a report on the state of the bridge describing the anomalies detected with respect to the results of the previous inspection is produced. All reports are searchable and accessible through the HMI application, in order to monitor and control both the bridge and the AVI system. Moreover, such reports provide the human inspector with the changes evaluation of the identified defects, hence assisting the final decision about the severity and—if needed—the actions to take.

### 2.1. Multi-Camera System

The *p.o.c* of the robot developed for the San Giorgio's bridge and presented in this paper hosts three different MCSs. The RGB camera selected is a Basler ace acA4112-8gc GigE [39], with a Sony global shutter IMX304 sensor that captures color images at 8 frames per second (fps) with a 12.3 megapixel (MP) (4096 × 3000 pixels) resolution. In order to capture a micro flaw on the bridge at an average distance of 0.8 m, an LM6FC f6.5 mm manual zoom video lens manufactured by Kowa [40] was mounted on the camera. The second camera for the development of the MCS is the FRAMOS depth camera D435E [41]. This device is composed of an active infrared (IR) and the RGB global shutter sensors in stereo system mode. The on-board sensor acquires depth and color images at 30 fps, with a resolution of 1280 × 720 pixels. Finally, the vision system is completed with the multispectral Ximea SM4 × 4-VIS3 camera [42]. This multispectral camera mounts a 4 × 4 snapshot-mosaic sensor (16 interference filters) in the spectral range from 460 nm to 600 nm, with a native resolution of 2048 × 1088 pixels, and spatial resolution of 512 × 272 pixels, with a temporal resolution of 170 fps. Figure 3 shows the multi-technology camera system with the RGB, 3D and multispectral cameras. In order to manage the different resolutions of the three cameras, cubic interpolation was applied; hence, the images match the resolution of the multispectral camera, which is the one with the lowest resolution. Nevertheless, images are also stored in the raw format to perform also offline analysis.

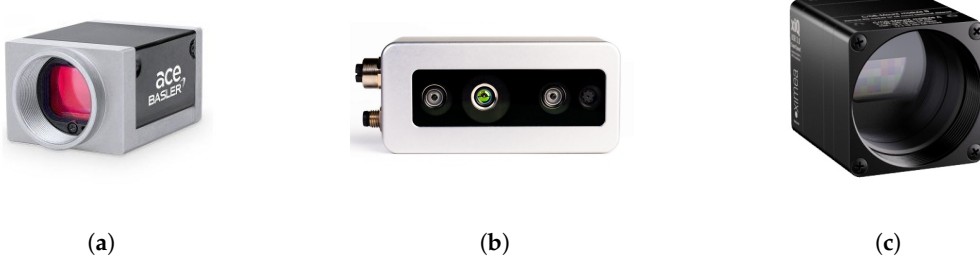

|    (**a**)    |    (**b**)    |    (**c**)    |

**Figure 3.** The multi-camera system composed of (**a**) RGB, (**b**) 3D, and (**c**) multispectral cameras.

The entire vision system has been enclosed in a special protective cover [43], in order to ensure IP67 standard (i.e., high resistance to dust and water). Such a feature is necessary to guarantee a correct functioning of the MCS outdoor in all weather conditions. Moreover, in order to produce high-quality results in different light conditions, two high-efficiency LED strips were mounted over the vision system. These strips are driven by a relay actuator and General-Purpose Input/Output (GPIO) module [44] to illuminate the scene when the camera shutter command is sent. Consequently, the robot overcomes the challenge of capturing images at night or in low-light conditions, and working during the night/day cycle.

Flaw Detection Algorithm

The data recorded by the MCSs are processed by a multi-step framework specifically designed to perform pattern analysis. Each camera of the MCS captures an image, and passes it to the visual inspection algorithm. In this first step, the Image Acquisition phase saves in 3D (i.e., $x$, $y$, and $z$ axes, respectively) the position of the robot beam, and the corresponding temporal metadata for fast flaw retrieval after the post analysis. Moreover, the position of the cameras is measured with respect to the sliding beam holding the MCS. A schematic representation of the framework developed for pattern analysis is depicted in Figure 4. The multispectral is a single channel, while RGB and 3D images are converted to grayscale to reduce processing time, also enabling the performance of the Contour Search and Hierarchy steps. This phase reduces the information of the images into a single channel, hence resulting in a gain in terms of processing time. Similarly, scaling down the resolution of both 3D and RGB images to the small size of the multispectral camera benefits the processing time. The triplet of grayscale images are then passed to the Blurring Filter. This filter implements the Gaussian blur (i.e., a nonuniform low-pass filter) that preserves low spatial frequency and reduces noise and negligible details. From a mathematical point of view, the Gaussian Filter convolves an image with a Gaussian kernel. Optionally, a Bilateral Filter with logarithmic transformation can be used. The Gaussian and the Bilateral filters share the same working principle, by scaling the values of the neighbours of a specific central pixel. The Gaussian filter mitigates the intensity values of the neighbours proportionally to the distance measured in terms of pixel coordinates. Conversely, the Bilateral Filter mitigates the neighbours proportionally to the variations of intensity with respect to the central pixel. Hence, the latter removes noise in flat areas, and avoids blurring the imaged objects, hence preserving high-frequency details.

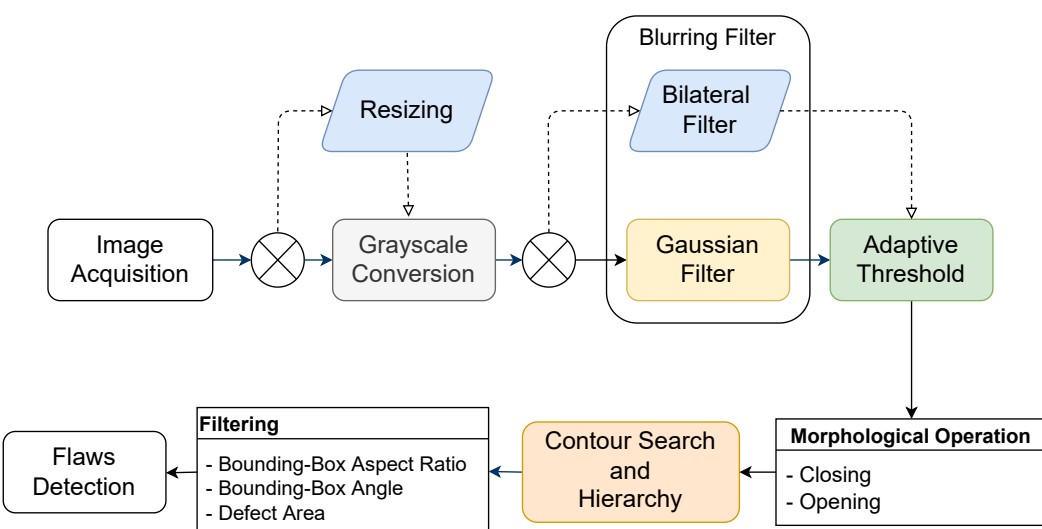

**Figure 4.** Schematic representation of the workflow of the proposed system to perform pattern analysis in the bridge inspection process.

In cascade to the image filtering, the Adaptive Threshold algorithm is performed in order to separate the objects of interests (e.g., defects) from the background, according to strong variations in the pixel intensity. Furthermore, nonlinear Morphological Operations are used to enhance the images, by using a squared kernel 9 pixels wide. In particular, Morphological Opening is an erosion process followed by dilation that removes small objects, while Morphological Closing is a dilation process followed by an erosion that is used to remove small holes. Moreover, the object Contour Search is performed. This function detects objects in an image and states the Hierarchy relationship between them. In particular, the Contour Search algorithm identifies the contours of the objects in the images, and evaluates if some contours are inside some others. The inner objects are

named children, while the outer ones are named parents. All the identified contours are further processed by using a set of three filters in order to remove possible outliers: (i) Bounding-Box Aspect Ratio, (ii) Bounding-Box Angle and (iii) Defect Area. The output of processing pipeline is a set of regions / bounding boxes of objects that represent potential defects. Bounding boxes and related parameters (e.g., area, orientation and aspect ratio) are then used to provide a classification (e.g., flaws). In addition, binary images are generated to highlight the detected flaws. These black and white images maps the pixels belonging to a specific flaw to "1", and the remaining pixels to "0". Selecting an appropriate threshold intensity is a critical step. If the threshold intensity is too high, flaws will not be detected. Conversely, if it is too low, the image becomes noisy and it is difficult to differentiate flaws from noise.

### 2.2. Robot Inspection Unit

RIU is a robotic structure able to autonomously scan the lower deck surface of a bridge (Figure 5). It has 3 degrees of freedom (DOFs), as it is able to move along the $z$ direction, i.e., parallel to the longitudinal direction of the bridge, the $x$ direction, i.e., parallel to the cross section direction of the bridge, and the $y$ direction, i.e., perpendicular to the lower deck surface of the girder. The first two DOFs are always active, while the third one may depend on the application. The motion along the $z$ direction is enabled by the two rails installed on the bridge, the one along the $x$ direction is given by the retractable beam the robot, while the $y$ direction is inspected by an add-on mechanism used to perform close or contact inspections.

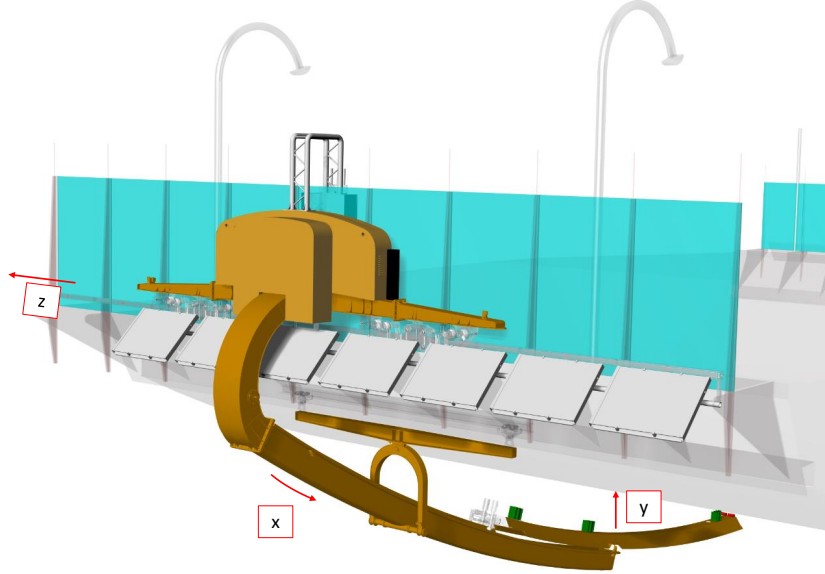

**Figure 5.** Representation of Robot Inspection Unit (RIU), with the three directions of the motion labeled with red boxes.

The RIU is composed of several sub-assemblies made of different materials, such as carbon fiber, steel and aluminum. *Fixed Beam* and *Moving Beam* of the robot were built in carbon fiber [45], in order to reduce the weight and the vibrations resulting from the motion and the wind hitting the surface. The set of mechanisms is also composed of wheel carriers and actuation systems that were manufactured in steel in order to guarantee high performance with respect to the cyclic loads. The *Upper Hanger* and *Lower Hanger* are the two largest welded components of RIU and were built in aluminum in order to reduce the overall weight of the whole system.

RIU is connected to the two rails through the wheels carriers that in turn are connected to the *Upper Hanger* and *Lower Hanger*. The wheels carriers rolling on the upper rail are actuated, while the ones rolling on the lower rail are towed with an isostatic assembly. The

*Upper Hanger* and *Lower Hanger* are connected through the *Fixed Beam*, which is bolted to the *Upper Hanger* structure, and pinned to *Lower Hanger*. The *Moving Beam* can move along the $x$ direction, by sliding over the *Fixed Beam*. The top surface of *Moving Beam* is equipped with three technological platforms that compose the multi-camera system, as described in Section 2. Furthermore, RIU is equipped with a set of sensors that makes it fully energetically autonomous, intelligent and correctly located with respect to the bridge geometry. The *autonomy* is guaranteed by an on board set of commercial batteries that are charged during the inspection. In particular, the bridge hosts several charging stations, enabling the robot to recharge over the inspection. The robot uses batteries with a lightweight design, with a very short charging time. Furthermore, these batteries perform consistently in a wide temperature range, spanning from $-40\,°C$ to $+60\,°C$, hence also enabling their use in outdoor environments (such as a bridge). The *intelligence* is given by the presence of a commercial ultrasound anemometer that estimates the suitability of the environmental conditions. This ultrasound sensor is able to detect wind speed compared to the traditional counterpart, and it requires less energy to work, thus making its use in "remote sites" suitable. In the case the wind speed exceeds a predefined upper bound, the robot reaches a safe position or assumes a safe configuration. This scenario, of course, is possible not only thanks to the anemometer, but also to a proper control system. It is capable of detecting the "not safe" conditions, and to properly drive the robot. Lastly, the *localization* with respect to the bridge geometry was obtained through a proper set of commercial distance lasers and encoders. Thus, the robot is self-aware of its position on the bridge, which is expressed in terms of the distance traveled in $z$ and in $x$ directions. The distance values measured by these two main groups of sensors have to match; otherwise, an input error is sent to the control system and the robot goes into a safe configuration.

*2.3. Data Structure and Processing*

The Wi-Fi network suitable to control the position of the robot and to transfer the image data is shown in Figure 6. The network is composed of three main cores: (i) PC Robot Server, (ii) PLC Master Robot and (iii) RIUs. All the internal communications among the three main cores take place through the Siemens proprietary bus, named PROFIBUS [46]. PROFIBUS is a protocol for field bus communication in automation technology that links controller or control systems with decentralized field devices (i.e., sensors and actuators) on the field level via a single bus cable. Moreover, it enables consistent data exchange with higher ranking communication systems. There are two types of PROFIBUS: Decentralized Peripherals (DP), when the Programmable Logic Controller (PLC) communicates with a field device (sensors and actuators), and Process Automation (PA), used to interface measuring instruments through a process control system. PROFIBUS has a very fast filed bus (12 Mbps) and it can work on the master-slave scheme by using token passing mechanism. A further reason for using this protocol is that its communication is specified in IEC 61158 and IEC 61784 (i.e., the international standard of communication), hence meeting the requirements for having a general purpose and scalable system. The communications between the PC Robot Server and the master-slave robot are sent via the Wi-Fi network. It uses a Siemens router device, which supports 2.4 GHz and 5 GHz transmissions with a bit rate up to 1300 Mbit/s. From the PC Robot Server via Wi-Fi connection, the movement commands are sent to the robots and the status and position of each robot on the bridge is monitored. The command arrives via Wi-Fi to the single robot, while the PLC Master Robot processes and sends the commands to all motor control drives, monitoring movement, control and emergency sensors. In the same way, the images captured by the multi-vision system are sent to the PC Robot Server to be processed with the flaw detection algorithm. Each acquired image file is composed of the image itself, and the metadata that contain additional information about (i) the relative position of the image acquisition ($x$, $y$ and $z$ axes position), (ii) the results of image processing pipeline with the defect severity level and (iii) the data from environmental sensors placed on the retractable beam. MQTTS and

HTTPS protocols are used to send a JSON object containing these structured data. Each acquisition stack contains about 40 MB of data.

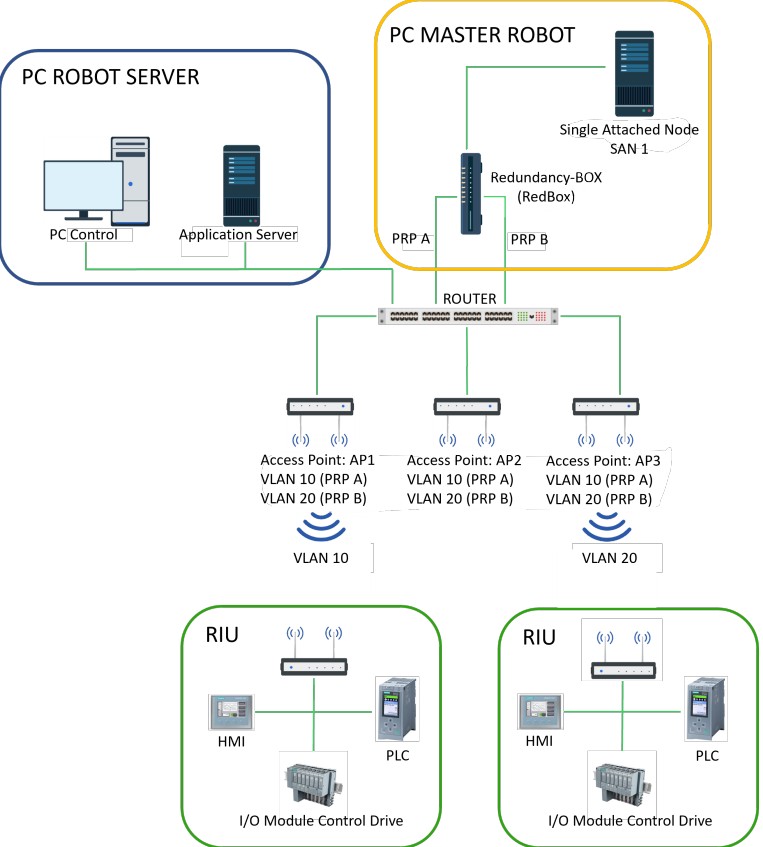

**Figure 6.** Schematic representation of the designed architecture of the Wi-FI network system.

*2.4. Control Software System*

The PC Robot Server is based on Windows Server. Two services will be running permanently: (i) a service for the communication between PLC Master and software SCADA and (ii) a web server for the collection of the data from the PLC Master and the computer vision system on the robot, which transfers them to the monitoring and control software system. The software system overview is depicted in Figure 7. The first service exposes to the SCADA system the data coming from the PLC Master and activates a limited set of functions. This service acts as an Open Platform Communications/Unified Architecture (OPC/UA) client connected to the PLC Master and as a Modbus TCP server to the SCADA system, allowing the latter to have controlled access to the PLC. The expected functionalities are: (i) read access to the state of a set of robot OPC/UA variables and (ii) executing start/stop commands for scheduled inspections. The state of the variable values is constantly updated through the OPC/UA subscription mechanism. Unfortunately, given the large number of variables present in the PLC Master, not all of them can be updated with the same frequency. Consequently, the priority is given to those variables considered critical for the system, with less frequent updates for the ones of secondary relevance. In order to meet the Modbus TCP specification, a set of registers was defined to which data from the PLC are published. The data in the registers are updated by OPC/UA subscriptions to the variables exposed by the PLC.

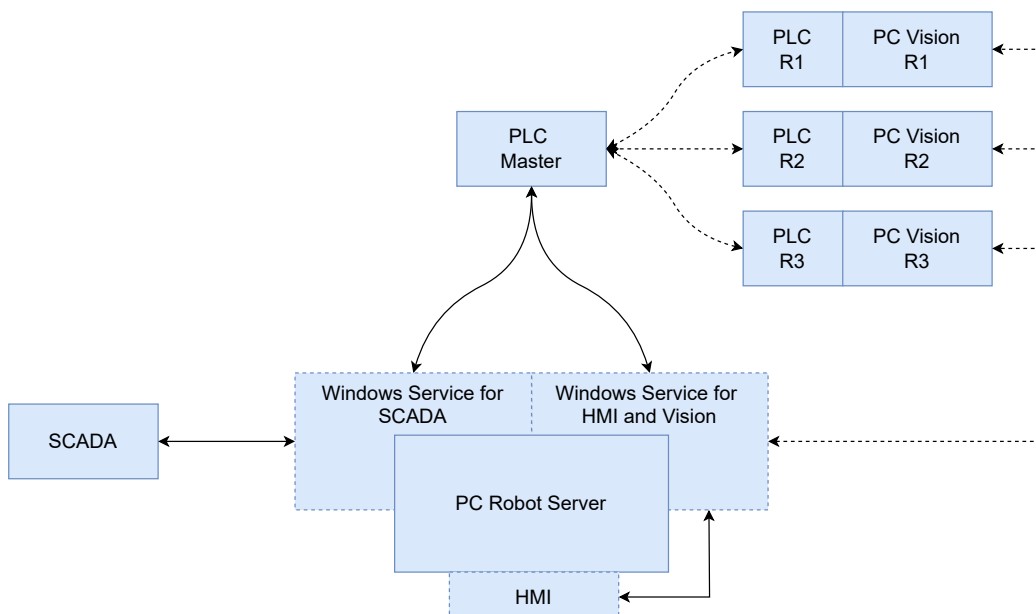

**Figure 7.** Schematic representation of the architecture of the software system.

Regarding the second service, the robot monitoring and control software are bootable from the PC Robot Server itself, producing an HMI. The web server provides authenticated APIs for accessing data from the PLC Master and collecting information from the computer vision systems in the various robots. This service collects data from the PLC Master via OPC/UA subscriptions, makes it available to the robot monitoring and controls the HMI application. Moreover, it makes it possible to execute OPC/UA methods on the PLC Master for robot control. The functionality of this service are: (i) read access to the status of robot variables exposed by the PLC Master, (ii) start/stop functionality of scheduled inspections, and (iii) control of individual robots as provided by the PLC Master. Similarly to the SCADA service, the status of the variables is updated through the OPC/UA subscription mechanism. The variables are divided into three different groups: General (for variables on the general state of the system), Robot Inspection North and Robot Inspection South. Furthermore, a set of authenticated REST APIs is used for communication with the computer vision systems connected to each robot. The functionalities guaranteed by the REST API consist of uploading in the PC Robot Server the photos acquired by the cameras connected to the robot, sending the results of the analysis of the acquired images to the PC Robot Server, reading the current position of the robot from the PC Vision and periodically sending the status of the PC Vision to the PC Robot Server.

The HMI provides the user with an integrated interface to monitor and control the real-time information related to the robot status obtained from the PLC Master and the vision systems. The user can start the programmed inspections, control each robot (i.e., master-slave) and give access to archived inspections. Interactive 3D models of the bridge and the robots are displayed in HMI, with the possibility to access a series of information, such as the status of the sensors, the captured photos and any detected anomalies or malfunctions. The application developed with web technologies is accessible via browser from the PC of the Robot Server itself. Each inspection robot has on board a PC to which all the cameras are connected, which performs image analysis when required. The PC receives the commands to start the photo capturing procedure from the robot PLC, performs its analysis and sends photos and results to the PC Robot Server. It also periodically sends its status in order to detect potential malfunctions, and obtains from the PC Robot Server information about the robot status useful to the computer vision system. The communication between the PC Vision on board the robots and the PC Robot Server has been implemented through the bidirectional channel MQTTS. A *client-server* model infrastructure has been realized, where PC Vision acts as a *client*, while PC Robot Server acts as a *server*. The PC Robot Server

waits for client requests, and as soon as they arrive, it provides each PC Vision with the configurations parameters and information related to the status of the robot. In turn, the PC Vision sends information to the PC Robot Server about its status, the values detected by the sensors and the events triggered by the image acquisitions. On the other hand, the images acquired during the inspections are not be exchanged on the bidirectional channel, but downloaded on request of the PC Robot Server. Information sent from PC Robot Server to PC Vision includes:

- Real-time updates on the status of the robot;
- Current position of the robot along the bridge (*z*-axis);
- Current position of the beam (*x*-axis), robot inspection only;
- Current position of the additional sensor (*y*-axis), robot inspection only;
- Values of the distance sensors on the robot;
- Charge level of the robot batteries;
- Current status and position of the inspection cycle, if it is running;
- Possible error and warning states of the robot;
- Any activation of the robot emergency stop;
- General error status of the system;
- Any configuration parameters of PC Vision. They are sent at the moment of connection and at the request of PC Vision

Conversely, information sent from PC Vision to PC Robot Server includes:

- Real-time updates on PC Vision status, such as possible camera and sensor malfunction status or possible general error status of the PC;
- The readings of the connected sensors (such as the ultrasonic sensor);
- All the events triggered by the acquisition, including the metadata that will be useful later to enrich the event (such as any URIs to find the photographs captured).

Finally, the PC Vision exposes a minimal web server where the acquired images are published, available to the PC Robot Server. The PC Robot Server implements a scheduled mechanism to copy the data present on the various PC Vision.

## 3. Results and Discussion

### 3.1. Flaw Detection Algorithm Results

The flaw detection algorithm was tested on RGB and 3D images, and the consequent results are reported in Figures 8 and 9, respectively. In particular, Figure 8 shows three different RGB images captured in different weather conditions: sunny (Figure 8a), night (Figure 8b) and wet (Figure 8c). These images are further processed by the flaw detection algorithm, and the consequent results are shown in the second column of Figure 8. The detected flaws are encapsulated in the colored bounding boxes. Finally, to highlight the flaw defects, binary images (Figure 8c,f,i) are generated from the processed images. These images will be useful in the decision making process, in order to validate the goodness of the identification, and the following actions are to be taken. Figure 8a,g show a portion of the lower deck surface, where welds were used to join the different metal parts. Processing these images shows several bounding boxes of different sizes, resulting from the size of the kernels of the related filters defined for the flaw detection algorithm. These flaws are considered as false negatives in the final decision-making step. Conversely, Figure 8d shows an artificial defect (marker stroke) and a small hole. The analysis of this image shows that the flaw detection algorithm is robust to color differences (even if artificial) and to the detection of small defects.

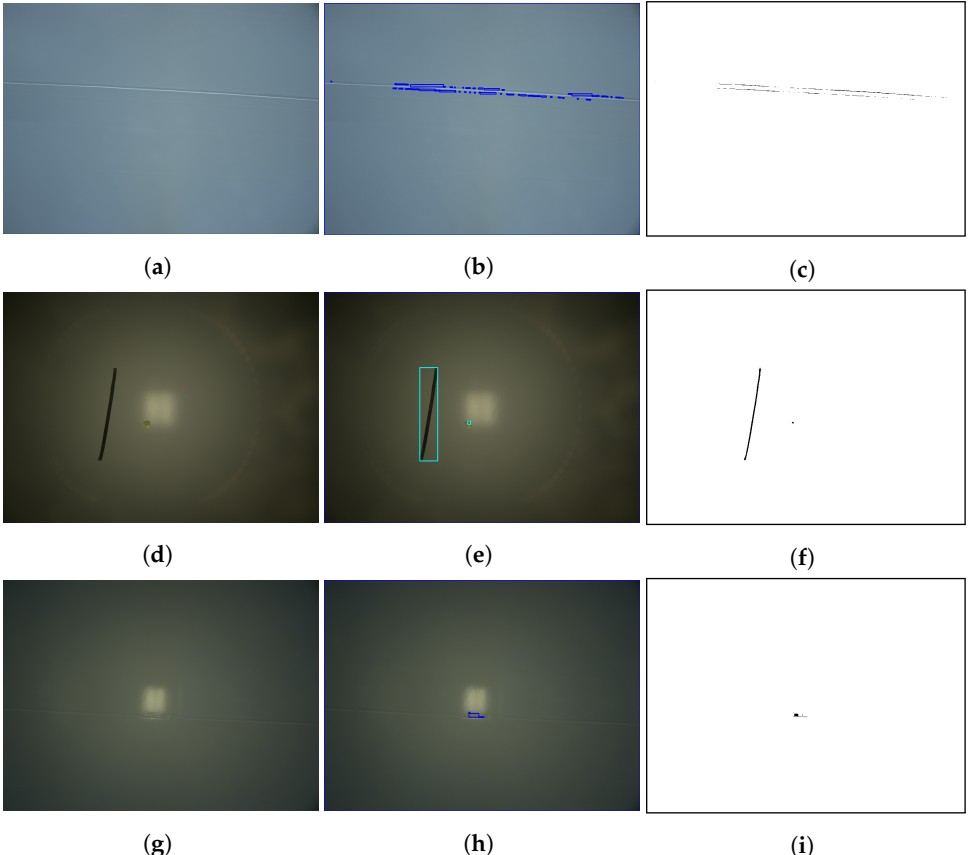

**Figure 8.** RGB images of part of the bridge under different weather conditions: sunny (**a**), night (**d**) and wet (**g**). Processed images by the flaw detection algorithm (**b**,**e**,**h**) show the colored bounding box encapsulates the flaws detected. Detected flaws are highlighted by binary images (**c**,**f**,**i**).

Figure 9 shows a 2D projection of a 3D image. The result of the flaw detection algorithm plots circular and rectangular bounding boxes that surround the flaws. The algorithm identified defects such as holes, bumps and color variation.

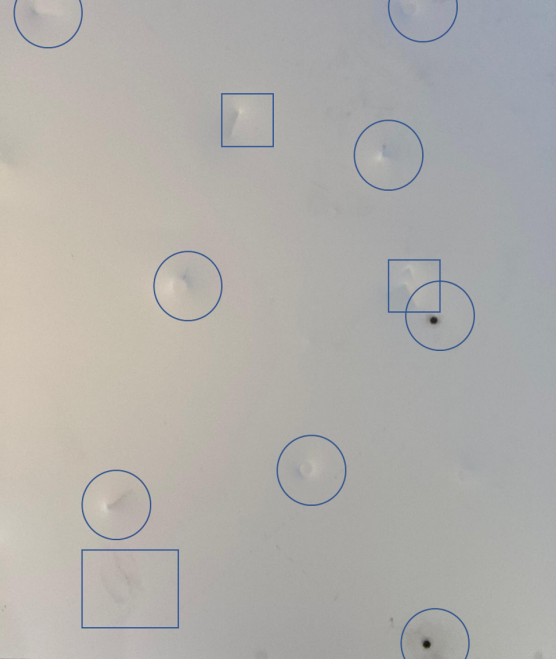

**Figure 9.** Result of the flaw detection algorithm performed on a 3D image.

The acquisition and processing phases of each image triplet (i.e., RGB, 3D and multi-spectral images) take less than 15 s that is an hard requirement. Such value was measured while performing the inspection process on the onboard Hystou computer [47] with the following features: Intel i7-5500U processor, 8 GB RAM and 1 TB SSD with Ubuntu 20.40LTS.

### 3.2. Robots Performance

In this section, the preliminary tests run in order to evaluate the RI performances are presented by analysing an example of the operative cycle. In this case, the robot moves back and forth along the $x$ axis three consecutive times, changing its $z$ motion in every iteration, in order to analyse different portions of the bridge. The actions performed by the robot are reported in Algorithm 1.

---

**Algorithm 1** RI performance example cycle

---

Starting position ($x = 0$ mm, $z = 0$ mm)
Movement along the $x$ axis ($x = +5800$ mm), with $z$ constant at 0 mm
Image capture
Movement along the $x$ axis ($x = -5800$ mm), with $z$ constant at 0 mm
Movement along the $z$ axis ($z = +600$ mm), with $x$ constant at 0 mm
Movement along the $x$ axis ($x = +5800$ mm), with $z$ constant at 600 mm
Pause for 20 s for vibration stabilization
Image capture
Movement along the $x$ axis ($x = -5800$ mm), with $z$ constant at 600 mm
Movement along the $z$ axis ($z = +2400$ mm), with $x$ constant at 0 mm
Pause for 10 s for vibration stabilization
Movement along the $x$ axis ($x = +5800$ mm), with $z$ constant at 2400 mm
Image capture
Movement along the $x$ axis ($x = -5800$ mm), with $z$ constant at 2400 mm
Movement along the $z$ axis ($z = +2400$ mm), with $x$ constant at 0 mm
Final position ($x = 0$ mm, $z = 0$ mm)

---

The trajectories along both the $x$ and the $z$ axes obtained while performing Algorithm 1 are depicted in Figure 10, represented in blue and orange, respectively.

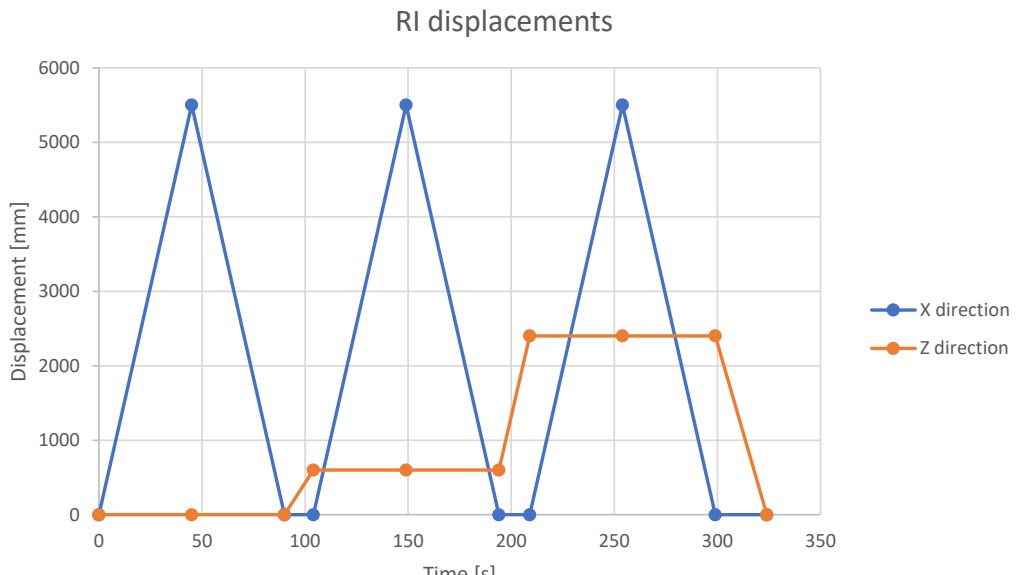

**Figure 10.** Schematic representation of the trajectories of the robot along the $x$ (in blue) and the $z$ (in orange) axes.

The cycle modeled by Algorithm 1 was designed as modular, ensuring the flexibility to adapt the motion of the robots to the needs. In particular, the movements can be reversed, and the distances can be modified. Unfortunately, in the specific case of the *p.o.c.* of the robot presented in this paper, the motion along the $x$ axis is limited to 4500mm, due to the physical length of the prototype itself. Conversely, the $z$ axis has no limitations; hence, the trajectory can be freely decided according the desired path.

Furthermore, the consumption level of the battery of the robot has been monitored and the results are displayed in Figure 11. The graph shows a linear dependency of the consumption level on the elapsing time.

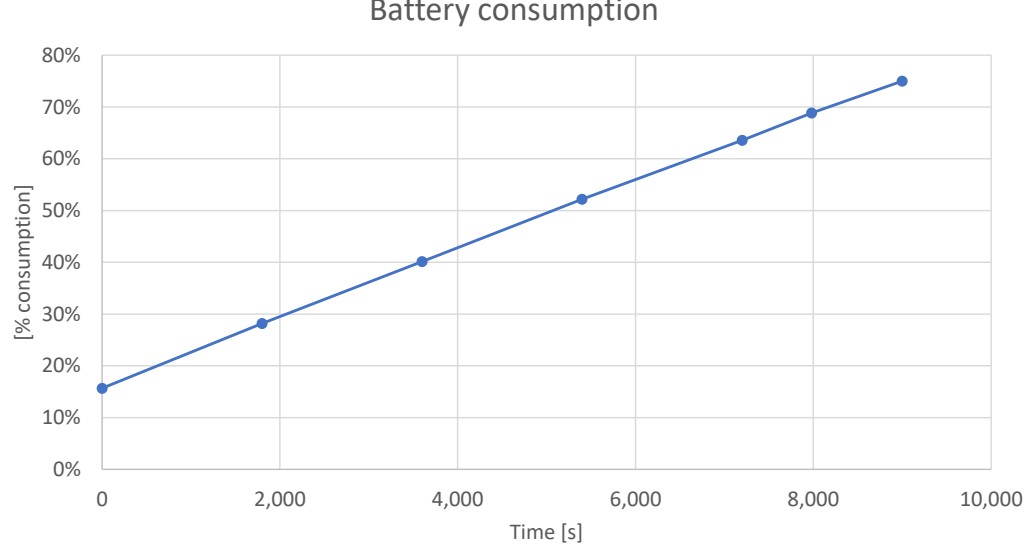

**Figure 11.** Schematic representation of the performance of the batteries consumption. The consumption level shows a linear dependency to the elapsing time.

The test associated with the results in Figure 11 was conducted for 9000 s (i.e., $\sim$2.5 h), in which the RIU performed inspection of the underside of the bridge. At the end of this test, the average battery consumption resulted to be equal to the 75% of the total capacity. The remaining 25% of the battery capacity was saved in order to make sure the RIU can get to the nearest charging station and to account for possible different needs. Moreover, such energy saving paves the way to further development of the robots, including additional sensors to improve the accuracy of the inspection.

## 4. Conclusions

Monitoring the health of bridges is a pivotal element for scheduling the maintenance operations, and ensuring the safety of the people passing close to or working on a bridge. Unfortunately, traditional bridge inspection techniques struggle to achieve a well-rounded inspection and to have ongoing monitoring of bridge health.

In this paper, a remote visual inspection system for bridge predictive maintenance is proposed, integrating robotics and vision solutions. The design of the inspection robot was not tailored on a specific application, but it actually was realised in a modular and flexible fashion. In fact, the length of the *Moving Beam*, the amount of multispectral vision systems and the Wi-Fi network can be declined to the different applications. The *p.o.c.* illustrated in this paper was designed for the San Giorgio bridge in Genova and is represented by a robotic platform, named RIU, able to scan the lower deck surface of a bridge by moving continuously from beginning to end along the longitudinal direction of the infrastructure. The robot is equipped with several different sensors such as ultrasound anemometer, lasers and encoders together with a multispectral vision system. The set of sensors on the robot is used to track the location of the robot along the bridge, while the vision system to

perform flaws detection on RGB, 3D and multispectral images. The use of single channel images ensures compliance with the manufacturer time limit of 15 s for the acquisition and processing phases of each image triplet. The design of RIU accounts for the presence of multiple vision systems, and the *p.o.c.* discussed in this paper hosts three replicas, which are installed in different strategic locations of the *Moving Beam*. The presence of several vision systems on the same robotic platform dramatically reduces the time required to scan the whole bridge. Moreover, the presence of a *Moving Beam* with its capability of moving along the transverse direction of the bridge makes the robot adaptable to bridges of different size and shape, or other type of "longitudinal" infrastructure (such as tunnels and chimneys). Additionally, the robot is equipped with a modular and scalable Wi-Fi network designed to handle data traffic, and to send the commands to all the motor control drives. Furthermore, the robot mounts a set of batteries that guarantees power autonomy while performing the scanning operations.

The flaw detection system proposed in this paper was proven to produce high-quality and robust results. However, the identification task is still an open issue in the field of computer vision, and the best performing ML and DL methods requires an incredibly large amount of real data, which are often difficult to obtain. Consequently, the data recorded by the developed *p.o.c.* will be used in the future to continuously update the system, hence guaranteeing better and better performance. Furthermore, in the near future, the possibility of using satellite imagery as an additional source of data will be considered. This kind of image allows measurement results to be obtained over a long period, sometimes over decades. These images also enable the detection of slow movements of various structures, as well as the monitoring of the surrounding environment. In parallel, in the near future, Generative Adversarial Networks (GAN) will be tested, exploiting their capabilities to produce good results even in the case of partial amount of data [48]. Additionally, further analysis will focus on the improvement of electrical efficiency, in order to reduce consumption and achieve an AVI system with *quasi*-zero impact on the environment.

The designed robotic system produces the same high-quality performance regardless of day or night or weather conditions. Moreover, it produces repeatable and consistent results, hence improving the objectivity of the inspection and easing the dependence on the human operators. Nevertheless, automating the identification process does not aim at removing the role of human inspectors, but actually at facilitating their decision-making process and increasing their safety. Moreover, the adoption of autonomous robots significantly decreases the amount of time required to perform an inspection, and hence, the necessary financial resources, both in terms of manpower and funds. It clearly emerges how such technology can boost the inspection field, with robotic systems that may be easily installed on a wide variety of infrastructures, continuously monitoring the health status, and thus, preventing serious damages. Similar scenarios would remarkably and positively affect the daily life of the society, and above all, the safety conditions of many people.

**Author Contributions:** Conceptualization, A.G. and M.D.; methodology, A.G., M.D. and A.M.; software, A.M. and M.S. (Michele Sasso); validation, A.G., M.D., A.M. and F.C.; formal analysis, A.G., M.D. and G.M.; investigation, A.G. and M.D.; resources, E.F. and F.C.; data curation, M.D. and A.M.; writing—original draft preparation, A.G., M.D. and G.M.; writing—review and editing, A.G., M.D., A.M., M.S. (Massimiliano Scaccia) and G.M.; visualization, A.M., M.S. (Michele Sasso) and G.M.; supervision, A.M.; project administration, E.F. and F.C.; funding acquisition, F.C. All authors have read and agreed to the published version of the manuscript.

**Funding:** This research received no external funding.

**Acknowledgments:** The authors would like to thank Camozzi, SDA, for their valuable contribution in supporting the research on new technologies to install on robots and to test them on the physical prototype to assess their potential.

**Conflicts of Interest:** The authors declare no conflict of interest.

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
