# Peer review of "A Novel Remote Visual Inspection System for Bridge Predictive Maintenance"

_remotesensing, doi:10.3390/rs14092248_

Round 1

Reviewer 1 Report

The article uses the term Unmanned Aircraft Systems (UASs), which is better replaced by the Unmanned Aerial Vehicle (UAV), commonly known as a drone, which is generally accepted in the scientific literature. Moreover, the abbreviation UAS is not used anywhere else in the text. In this sense, it is redundant.
Fig. 8 and 9 are not presented in the text, which makes it impossible to evaluate the effectiveness of the proposed algorithms.
Wish to the authors: to consider the possibility of using satellite data as an additional channel of information. Despite the lower resolution of satellite images, they make it possible to obtain measurement results over a long period, sometimes over decades. This makes it possible to detect slow displacements of various structures and prevent their destruction.
For bridge structures located in the mountains, it is important to prevent avalanches and mudflows that can destroy them. Satellite measurements can also be useful for these tasks.
Absence Fig. 8 and 9 does not allow to evaluate the article and publish it in its current form.

Author Response

The authors would like to thank the Reviewer for his valuable comments that have helped us to improve significantly the quality of our submitted article.

The article uses the term Unmanned Aircraft Systems (UASs), which is better replaced by the Unmanned Aerial Vehicle (UAV), commonly known as a drone, which is generally accepted in the scientific literature. Moreover, the abbreviation UAS is not used anywhere else in the text. In this sense, it is redundant.

> Thank you for this comment. We have replaced Unmanned Aircraft Systems (UASs) with Unmanned Aerial Vehicle (row 45).

Fig. 8 and 9 are not presented in the text, which makes it impossible to evaluate the effectiveness of the proposed algorithms.
Wish to the authors: to consider the possibility of using satellite data as an additional channel of information. Despite the lower resolution of satellite images, they make it possible to obtain measurement results over a long period, sometimes over decades. This makes it possible to detect slow displacements of various structures and prevent their destruction.
For bridge structures located in the mountains, it is important to prevent avalanches and mudflows that can destroy them. Satellite measurements can also be useful for these tasks.
Absence Fig. 8 and 9 does not allow to evaluate the article and publish it in its current form.

> We apologize to the reviewer for the inconvenience. Due to upload problems, Figs 8 and 9 were not uploaded correctly. Now, both figures are displayed correctly.  Concerning satellite imagery, we mentioned them in the Conclusion section to describe future work (rows 479-483).

Reviewer 2 Report

In this work the authors developed a unique new concept for predictive maintenance on infrastructures which goes beyond the traditional inspection methods that are not objective, tied to the inspector’s experience and require human presence on site. This method is relied on the remote visual inspection system to perform predictive maintenance on infrastructures such as bridges. This is based on the fusion between advanced robotic technologies and the Automated Visual Inspection that guarantees objective results, high-level of safety and low processing time of the results. . The flaw detection system proposed in this paper was proven to produce high-quality and robust results. However, some practical applications in order to prove the efficiency of the method should be performed in the near future.

The paper addresses a topic posing numerical and experimental challenges and having practical significance. It is methodologically correct. The paper is suitable for publication. Some comments to the authors are:

  • Figures 8 and 9 is not clear seen.
  • The literature review in introduction is thorough and it is very well written, however some additional reference listing below regarding the application of the wavelet analysis in damage detection of structure that can be applied in bridges could be added in the introduction chapter.
  1. Pnevmatikos, N.G. & Hatzigeorgiou, G.D. (2016), “Damage detection of frame structures subjected to earthquake excitation using discrete wavelet analysis”, Bulletin of Earthquake Engineering, 15(1), 227-248, DOI 10.1007/s10518-016-9962-z.
  2. Pnevmatikos, F. Konstandakopoulou, B. Blachowski, G. Papavasileiou and P. Broukos, (2020), ‘Multifractal analysis and wavelet leaders for structural damage detection of structures subjected to earthquake excitation’, Soil Dynamics and Earthquake Engineering, 139, https://doi.org/10.1016/j.soildyn.2020.106328.

Author Response

The authors would like to thank the Reviewer for his valuable comments that have helped us to improve significantly the quality of our submitted article.

In this work the authors developed a unique new concept for predictive maintenance on infrastructures which goes beyond the traditional inspection methods that are not objective, tied to the inspector’s experience and require human presence on site. This method is relied on the remote visual inspection system to perform predictive maintenance on infrastructures such as bridges. This is based on the fusion between advanced robotic technologies and the Automated Visual Inspection that guarantees objective results, high-level of safety and low processing time of the results. The flaw detection system proposed in this paper was proven to produce high-quality and robust results. However, some practical applications in order to prove the efficiency of the method should be performed in the near future.

> We thank the reviewer for the positive and constructive suggestions. The proposed method will continuously be refined to improve the results in the next future, as described in the manuscript. Additionally, we will evaluate several methods to reinforce and further validate our results.

The paper addresses a topic posing numerical and experimental challenges and having practical significance. It is methodologically correct. The paper is suitable for publication. Some comments to the authors are:

  • Figures 8 and 9 is not clear seen.

> We apologize to the reviewer for the inconvenience. Due to upload problems, Figs 8 and 9 were not uploaded correctly. Now, both figures are displayed correctly.

  • The literature review in introduction is thorough and it is very well written, however some additional reference listing below regarding the application of the wavelet analysis in damage detection of structure that can be applied in bridges could be added in the introduction chapter.
  1. Pnevmatikos, N.G. & Hatzigeorgiou, G.D. (2016), “Damage detection of frame structures subjected to earthquake excitation using discrete wavelet analysis”, Bulletin of Earthquake Engineering, 15(1), 227-248, DOI 10.1007/s10518-016-9962-z.
  2. Pnevmatikos, F. Konstandakopoulou, B. Blachowski, G. Papavasileiou and P. Broukos, (2020), ‘Multifractal analysis and wavelet leaders for structural damage detection of structures subjected to earthquake excitation’, Soil Dynamics and Earthquake Engineering, 139, https://doi.org/10.1016/j.soildyn.2020.106328.

> The authors thank the reviewer for the suggestions. The recommended references were added in the Introduction section at row 83.

Reviewer 3 Report

The paper has a clear, easy-to-read style with moderate English inaccuracies. I skip therefore language issues and concentrate on the content.

The most disturbing fact is the complete failure of the result images in Figure 8 and 9. The evaluation was written with the “hope” that they are acceptable.

The details are the followings:

  • line 45: I would not differ UASs and drones; nowadays, the literature handles them as synonyms.
  • line 102: I would write Scanning in capital when Terrestrial Laser Scanning term is used and abbreviated
  • line 104: Speaking about photogrammetry and TLS together, “image-based inspection” is not the proper term characterizing TLS measurements, so please choose another expression.
  • line 118: What is p.o.c stand for? Is it Proof-of-concept? Please use the unabbreviated form for the first time and introduce the abbreviation, and then later, the abbreviated form is OK.
  • line 120: Robot Inspection (RI) is used in the whole paper with two disturbing meanings. The first is the procedure: as the measurements, the inspections are executed. The second meaning is the hardware-software system, the unit, which has a material realization. To avoid these twofold meaning I suggest that the second case Robot Inspection Unit (RIU) or alternatively the inspection robot name can be written.
  • line 137: the developed system is composed of a multi-camera system (MCS) and a robotic platform. The illustration Fig. 2 contains Multi-Vision System. I suggest the use of the same expression (MCS).
  • line 161: what is AVI stand for? Automatic Visual Inspection? In this context, I couldn’t resolve it.
  • line 168: to have the chance to compare the system components, I suggest specifying the camera resolution also in pixel × pixel style, not only in megapixels.
  • line 179: after presenting the cameras, one important question isn’t answered: how is the resolution difference managed? Is an interpolation applied?
  • line 198: grayscale is used to have improved accuracy. Is it really true? Do we have to disclaim the color information? How can color changes be discovered if only grayscale images are handled? Much worse (line 201), the reason is to achieve processing time! If a bridge inspection methodology is being developed, real-time processing plays not a crucial role. Here, in the presented context, similarly processing time has not such a high importance, because color information can also be processed in the acceptable time frame.
  • line 201: downscaling the geometric resolution to achieve processing time – this is the same problem! If the goal is to detect cracks and similar flaws, and we have a 12.3 megapixel camera with a 0.1 megapixel (512×272) multispectral camera, it is really the right choice to have the lower resolution?! Grayscale image is also derived from multispectral camera image?! It has then relatively low sense…
  • lines 217 to 219: the definition of opening and closing can be found in image processing and computer vision books. The parametrization of the used morphological operation is in contrast, unique and exciting: type and size of the structuring element etc.
  • line 268: ultrasound sensor detects wind speed without underlining the lower limit – it’s clear, so I would delete the word “lower”
  • 4: what do represent vision components R1 to R4? I thought we have three vision sensors (RGB, RGBD, and multispectral), so why the 4th component? Or do I misunderstand something?
  • line 316: what is OPC/UA abbreviation? There is no explanation.
  • line 411: if the image acquisition and processing take ~15s, why do we have the limitation described before?! I haven’t read technical details about the control and processing computer hardware, as well as the used programming language and supporting libraries. Maybe it would improve the technical correctness of the paper.
  • 8: I can read only the placeholder names instead of the resulting images!
  • Algorithm 1: the term “picture” could be replaced with image capture, exposition, or similar terms
  • 9: the illustration fails completely here – only a placeholder is available
  • 11: the figure holds not too much information; the linearity is already in the text, so that I would skip this diagram.
  • line 465: ML and DL stand for machine/deep learning? Please solve the abbreviation!
  •  

Author Response

The authors would like to thank the Reviewers for their valuable comments that have helped us to improve significantly the quality of our submitted article.

The paper has a clear, easy-to-read style with moderate English inaccuracies. I skip therefore language issues and concentrate on the content.

The most disturbing fact is the complete failure of the result images in Figure 8 and 9. The evaluation was written with the “hope” that they are acceptable.

> We apologize to the reviewer for the inconvenience. Due to upload problems, Figs 8 and 9 were not uploaded correctly. Now, both figures are displayed correctly.

The details are the followings:

  • line 45: I would not differ UASs and drones; nowadays, the literature handles them as synonyms.

>  We agree with the reviewer that the comment improves the quality of the work, and it hence has been implemented in the document.

  • line 102: I would write Scanning in capital when Terrestrial Laser Scanning term is used and abbreviated

> We thank the reviewer for the suggestion. The noun has been written in capital letters.

  • line 104: Speaking about photogrammetry and TLS together, “image-based inspection” is not the proper term characterizing TLS measurements, so please choose another expression.

> We agree with the suggestion of the reviewer. The sentence has been modified in the text.

  • line 118: What is p.o.c stand for? Is it Proof-of-concept? Please use the unabbreviated form for the first time and introduce the abbreviation, and then later, the abbreviated form is OK.

> The authors thank the reviewer for highlighting the abbreviation error. It has been clarified in the text.

  • line 120: Robot Inspection (RI) is used in the whole paper with two disturbing meanings. The first is the procedure: as the measurements, the inspections are executed. The second meaning is the hardware-software system, the unit, which has a material realization. To avoid these twofold meaning I suggest that the second case Robot Inspection Unit (RIU) or alternatively the inspection robot name can be written.

> The authors agree with the reviewer that RI can be misleading. Hence the unit has been named RIU as suggested, and the text has been modified accordingly.

  • line 137: the developed system is composed of a multi-camera system (MCS) and a robotic platform. The illustration Fig. 2 contains Multi-Vision System. I suggest the use of the same expression (MCS).

> The expression “multi-vision system” has been replaced with MCS as suggested by the reviewer.

  • line 161: what is AVI stand for? Automatic Visual Inspection? In this context, I couldn’t resolve it.

>  We thank the reviewer for the suggestion. However, the AVI acronym has already been defined in row 54.

  • line 168: to have the chance to compare the system components, I suggest specifying the camera resolution also in pixel × pixel style, not only in megapixels.

> The authors appreciate the reviewer suggestion, and the camera resolution has been added to the text (row 172).

  • line 179: after presenting the cameras, one important question isn’t answered: how is the resolution difference managed? Is an interpolation applied?

> We thank the reviewer for raising the point. The images recorded by the cameras are interpolated (cubic interpolation) in order to match all the lowest resolution of the multispectral camera. The same description has been added in the text (rows 183-186).

  • line 198: grayscale is used to have improved accuracy. Is it really true? Do we have to disclaim the color information? How can color changes be discovered if only grayscale images are handled? Much worse (line 201), the reason is to achieve processing time! If a bridge inspection methodology is being developed, real-time processing plays not a crucial role. Here, in the presented context, similarly processing time has not such a high importance, because color information can also be processed in the acceptable time frame.

> The authors appreciate the comment of the reviewer and apologize for poorly phrasing such an important concept. Using colour images do not lower the accuracy of the identification process. However, colour changes are not dealt with in the system, hence colour data do not bring useful information. Therefore, RGB images have only the disadvantage to slow the processing time. The text has been modified accordingly in order to avoid misleading information.

  • line 201: downscaling the geometric resolution to achieve processing time – this is the same problem! If the goal is to detect cracks and similar flaws, and we have a 12.3 megapixel camera with a resolution of 512×272 multispectral camera, it is really the right choice to have the lower resolution?! Grayscale image is also derived from multispectral camera image?! It has then relatively low sense…

> We thank the reviewer for highlighting a not very clear passage in the text. The multispectral camera produces natively single channel image that is then post-processed to extract spectral information according to the 4x4 multi-spectral colour filter array pattern. The text has been modified in order to improve this concept.

  • lines 217 to 219: the definition of opening and closing can be found in image processing and computer vision books. The parametrization of the used morphological operation is in contrast, unique and exciting: type and size of the structuring element etc.

>  We apologize for the lack of details. We used a 9x9 kernel of square type for the morphological operation, and this data have been added to the text (row 223).

  • line 268: ultrasound sensor detects wind speed without underlining the lower limit – it’s clear, so I would delete the word “lower”

> We thank the reviewer for the suggestion. The text has been modified as suggested, by deleting the word “lower”.

  • 4: what do represent vision components R1 to R4? I thought we have three vision sensors (RGB, RGBD, and multispectral), so why the 4th component? Or do I misunderstand something?

> We appreciate the suggestion. We have corrected the figure inserting only 3 robots that represent the three replicas of RIU.

  • line 316: what is OPC/UA abbreviation? There is no explanation.

> The definition of OPC/UA (Open Platform Communications  and Unified Architecture) has been added to the text.

  • line 411: if the image acquisition and processing take ~15s, why do we have the limitation described before?! I haven’t read technical details about the control and processing computer hardware, as well as the used programming language and supporting libraries. Maybe it would improve the technical correctness of the paper.

>  The authors apologize for the lack of information on the processing time. The 15s limit was a requirement given by the manufacturer that identified in that value the trade-off between processing time and quality of the whole inspection process. Moreover, the hardware specifications of the working station used to perform the analysis in 15s have been added to the text, in agreement with the comment of the reviewer. We store also data in raw format to perform off-line analysis if needed without having constraints.

  • 8: I can read only the placeholder names instead of the resulting images!

> We apologize to the reviewer for the inconvenience. Due to upload problems, Figs 8 and 9 were not uploaded correctly. Now, both figures are displayed correctly.

  • Algorithm 1: the term “picture” could be replaced with image capture, exposition, or similar terms

> We agree with the reviewer. We changed “picture” with “image capture”.

  • 9: the illustration fails completely here – only a placeholder is available

> We apologize with the reviewer. The problem has already been solved as described in the previous comment.

  • 11: the figure holds not too much information; the linearity is already in the text, so that I would skip this diagram.

> Thank you for this suggestion, but to improve the readability and interpretation of the text we want to maintain this figure.

  • line 465: ML and DL stand for machine/deep learning? Please solve the abbreviation!

> We thank the reviewer for the suggestion. ML and DL stand for machine/deep learning. The acronyms have already been defined in row 74.

Round 2

Reviewer 1 Report

The authors took into account my comments. I have no more objections.